# SARS-CoV-2 mutations among minks show reduced lethality and infectivity to humans

**Tomokazu Konishi** [ID] *

Graduate School of Bioresource Sciences, Akita Prefectural University, Akita, Japan

* konishi@akita-pu.ac.jp

## Abstract

SARS-CoV-2 infection in minks has become a serious problem, as the virus may mutate and reinfect humans; some countries have decided to cull minks. Here, the virus sequencing data in minks were analysed and compared to those of human-virus. Although the mink-virus maintained the characteristics of human-virus, some variants rapidly mutated, adapting to minks. Some mink-derived variants infected humans, which accounted for 40% of the total SARS-CoV-2 cases in the Netherlands. These variants appear to be less lethal and infective compared to those in humans. Variants that have mutated further among minks were not found in humans. Such mink-viruses might be suitable for vaccination for humans, such as in the case of the smallpox virus, which is less infective and toxic to humans.

**Data Availability Statement:** Data are available from figshare: https://doi.org/10.6084/m9.figshare.13385192.v1.

**Funding:** The author(s) received no specific funding for this work.

## Introduction

Severe acute respiratory syndrome coronavirus 2 (SARS-CoV-2), the cause of coronavirus disease (COVID-19), infects not only humans but also several other animal species [1]. Infection in minks has become a particularly serious problem [2–5]; symptoms in minks appear to be lethal in the USA and Denmark [6, 7], but are milder in Spain [8]; those in the Netherland seem to have been varied [5]. Minks are culled because of the suspicion that the mink-virus can mutate and infect humans again [3, 4]. Here, we report the results of principal component analysis [9] of the mink- and human-virus sequences. Many of the mink-viruses were identical to that of humans; however, some variants mutated rapidly. One such variant that was closer to the human-SARS-CoV-2 variant (human-virus) was prevalent in humans of the Netherlands, amounting to approximately 40% of the total cases. This variant was probably less lethal. Other variants that have mutated further among minks are unlikely to infect humans. If mink farming continues, more variants that have a low affinity to humans will become available. It is possible that such mink-SARS-CoV-2 variants (mink-viruses) could be used for vaccination in humans, such as in the case of the smallpox virus, which is less infective and toxic to humans.

Mink livestock have been farmed for a long time as their fur has a commercial value [10]. Although mink farming has been declining recently, up to 50 million animals are farmed worldwide, mainly in Europe.

**Competing interests:** The authors have declared that no competing interests exist.

## Materials & methods

Principal component analysis (PCA) represents a sequence matrix, which is inherently multivariate data on multiple axes [11]. Each axis covers a certain set of base positions with specific weights. These are principal components (PCs) for the bases. A sample is given a value on each axis, PC for samples. PCs for bases and samples are inextricably linked to each other. The high-level axes, such as PC1, represent differences associated with more samples and bases; conversely, the lower axes represent a minor difference, for example, a feature that appears only in a particular country or region.

Sequence data of 1,832 human-virus in the Netherlands, 6,980 in Denmark, and mink-virus of 188 Netherlands and 63 Denmark were downloaded from GISAID [12] on December 2nd, 2020. Sequences from 17,571 European human-virus downloaded previously were also used. The list of samples and acknowledgments are available in Figshare [13]. The sequences were aligned using DECHIPER [14] then analysed using PCA [11].

The axes were identified using 103 mink-virus and 6092 human-virus that were proportionally selected from each continent. Assuming a global human population of 7.8 billion, a simple ratio calculation can be drawn by using the estimated population of minks $x$, 103: $x = 6092:7.8E9$. Hence, these data may be comparable to 130 million minks. Since the mink population is estimated as 50 million [10], the axes provide two to three times the weight in PCA toward minks, which might have enabled the identification of unique mutations in mink-viruses. Following this, the axes were applied to all data.

All calculations were performed using R [15]. All the codes used are presented in the author's page in Figshare [13]. The number of confirmed cases and deaths were obtained from the homepage of the WHO [16]. The rate of fatality was estimated as the rate between deaths in the following week and the number of cases during the week. The 95% and 50% confidence intervals of the rate were estimated using the binomial test [17], according to the estimation that deaths occur randomly among patients; if history repeats in the area with the same conditions, the observed rates will most likely vary as in this estimate. The mink-derived human-virus were found to have sPC8 > 0.003 and sPC9 > 0.001, and the rate was estimated within the countries. The weekly estimation of rates varied, as the numbers of infected and dead, and especially the number of sequences registered, were not very high. Therefore, the line representing the status changes was smoothened using the LOWESS function of the R [18].

## Results and discussion

The two highest axes of PC are shown in Fig 1A (for basics of the methodology, see Materials and Methods). As is apparent, viral variants in human and mink appeared into four groups that are temporarily numbered as 0–3. Mink-virus in the Netherlands consisted of all groups, whereas those in Denmark belonged only to group 1. The highly infective variants of the second wave [19] were not found. In the sequence data from both countries, variations in mink-virus that appeared in those axes were few, forming concentrated points stacked with each other (Fig 1A). The limited number of variations reflects that viral transmission from humans to minks is rare and that the migration of the virus is also rare. The contribution of the viral samples from minks to these axes was small, and the viral samples from minks appeared in exactly the same PC as one of the humans. The axes represent the process by which SARS--CoV-2 adapts to humans [19]; group 3 includes the earliest variants, group 0 includes the first variant transferred to Europe [20], and groups 1 and 2 were derived from group 0. The routes of virus migration and mutation presented in these axes were completed in April, and all groups could be found on all continents.

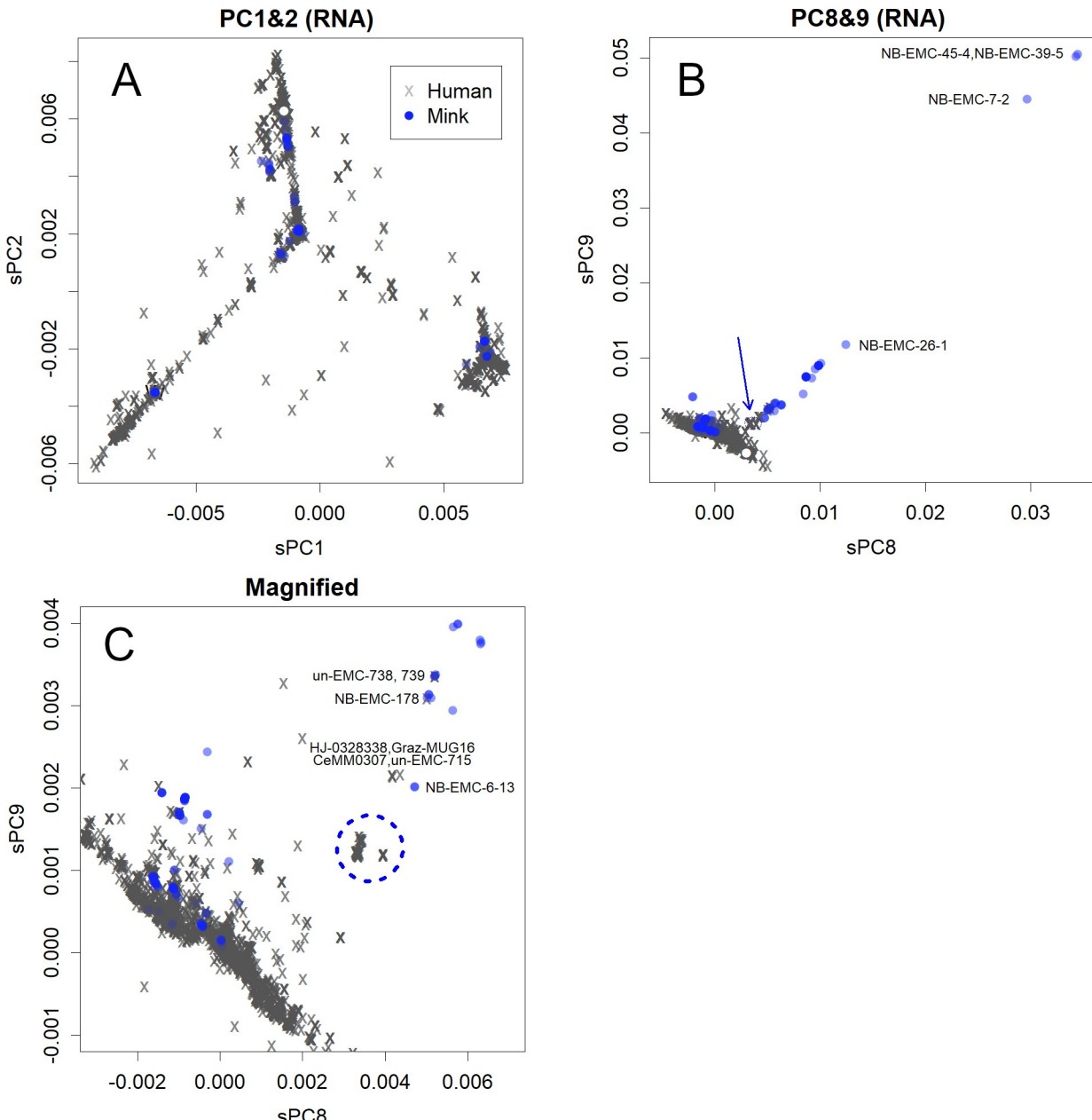

**Fig 1. Principal Component Analysis (PCA) for samples.** A. PC1 and PC2. These axes show the adaptation process of SARS-CoV-2 to humans [19]. Seven thousand of mink-virus are coloured in transparent blue; concentrated blue show multiple stackings. This is the same in 17,000 human-virus. The white dot indicates the position of NB-EMC-35-3, a group 2 mink-virus (S1 Table). **B**. PC8 and PC9. Mutations found in group 2 mink-virus in the Netherlands are presented on these axes. The presented IDs are those for mink-virus. The downward blue arrow indicates the mink-derived human-virus. **C**. Part of panel B enlarged. All IDs indicate human-virus, except for NB-EMC-6-13. The mink-derived human-virus are circled with a blue dotted line.

However, mutations also occurred in mink-virus in the Netherlands (Fig 1B). The directions of mutations among mink- and human-viruses are completely different from one another as shown in Fig 1A and 1B. As shown in Fig 1A, this difference suggests that the parental strain of SARS-CoV-2 did not originate from minks, otherwise the reversion mutation would return to the direction that occurs among humans. None of the bases or amino

acids were unique to mink-virus [5]. They can be recognised as characteristic rearrangements that have collected parts of human-virus. Such directed mutations were only found in group 2 variants of the Netherlands and were not found in mink-virus in Denmark.

These mutations may be the result of adaptation during viral transmission between minks, and the mutation rates were high (S1 Table) [5]. The differences are comparable to half of those between peaks of the seasonal H1N1 influenza virus, which usually takes a few years among humans to gain enough cumulative mutations that allow escaping herd immunity [19]. In addition, the rates of the missense mutations were high [21]. These phenomena were also observed in mutations in human-virus [19]. The direction strongly suggests that the mutations are meaningful rather than random. These properties could be due to the positive selection of variants that are more infectious among new hosts, minks.

Viruses that normally reside in humans can mutate and infect other mammals, such as mink. These viruses may be able to reinfect humans. Fig 1B and the enlarged insert (Fig 1C) depict the large population of human-viruses from which the mink-viruses evolved (Panel C). Many of the viruses have been reported only once. However, there were many infected humans in the group indicated by the dotted circles. Despite their large numbers, it is clear that these viruses do not normally dwell in humans. No direction of mutations that would enable human infection has been observed (Fig 1C). The viruses have not been reported in minks, which could reflect the very small number of reports from minks. However, based on their PCs it is reasonable to assume that they are derived from minks. The viruses were once prevalent mainly in the Netherlands (Fig 2A) and accounted for approximately 40% of all viruses present (Fig 2B). The variant may have been overlooked by the previous study [5]. Furthermore, human-to-human infectivity seemed to decrease and become sporadic (Fig 1C; the variants are identified).

Variants further away than those from human-viruses' mass have not yet been identified in humans (Fig 1B). Conversely, highly infective viral variants of the second wave in humans [19] have not yet been identified in minks. These human and mink variants could have adapted specifically to their respective hosts.

In many countries, the variants mutated before the second wave [19]. This phenomenon has also been confirmed in the Netherlands and Denmark, where similar variants were prevalent and waves occurred similarly (Fig 2 and S1 Fig). However, there is one difference; the variant that seems to be derived from minks appeared mainly in the Netherlands (Fig 2A and 2B). This variant has disappeared to be replaced by the variant belonging to group 0 that caused the second wave. This is also the case in Denmark (S1B Fig). This shows that the mink-derived variant is less infective than the second wave variant.

After the mink-derived variant disappeared, the fatality rate in the Netherlands increased (Fig 2B). In the first wave, many of the cases might have been ignored; hence, the rate became very high, and then decreased. The confidence intervals showed that the increase was substantial (Fig 2C). This difference may correspond to the proportion of the mink-derived human-virus present; if the variant is not lethal to humans, this difference can be explained by its disappearance. Actually, many mink farmers positive by PCR did not show heavy symptoms, and variants that related to the clusters of mink sequences found on the mink farms did not spill over to people living in close proximity to mink farms had occurred [5]. Since lower infectivity suggests a slow spread in the human body, it could allow sufficient time for the immune system to function, achieving lower lethality. Unfortunately, no data are available regarding individual human medical conditions; hence, we were unable to verify this lethality directly. However, data from 360 human samples with these mink-derived variants are shown in Figshare [13] to enable clinical verification. Additionally, to check the toxicity of other variants, PCs of other human samples from each continent are available on the same page in Figshare.

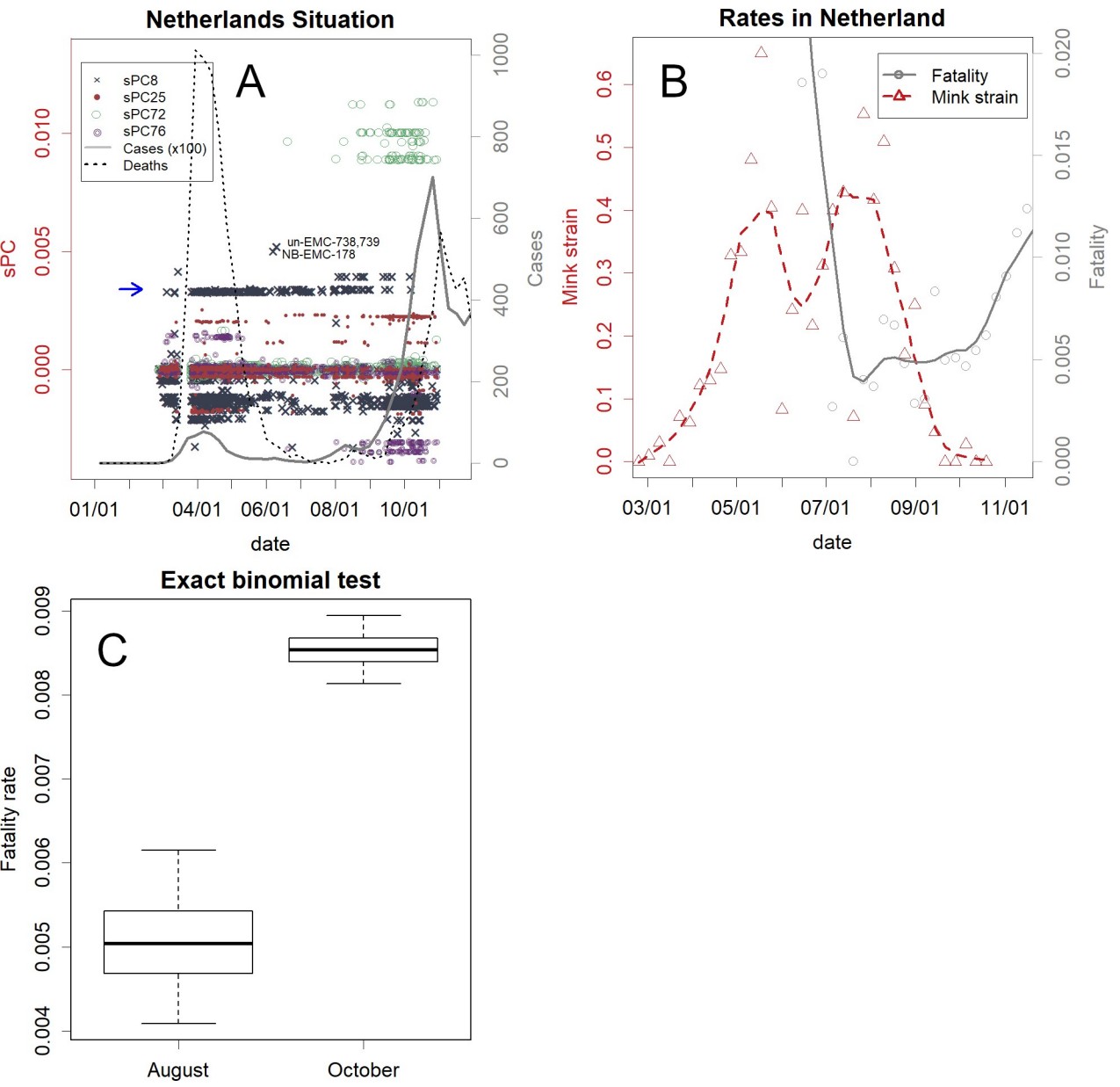

**Fig 2. The SARS-CoV-2 outbreak in the Netherlands. A**. Number of confirmed cases, deaths, and PC for human-virus. Before the second wave, variants had changed (coloured points of specified axes of PCs). The mink-derived human-virus appeared at and above the level of PC8 (x) indicated by the blue arrow. **B**. Rates of fatality (number of deaths in the following week/cases, grey) and the mink-derived variants (red). **C**. Comparison of fatality rates in August and October. The thick horizontal line indicates the estimation of the rate. The whiskers show 95% confidence intervals, and the boxes show the quartiles. The situation in Denmark is presented in S1 Fig.

Low lethality can also be expected, as the virus was derived from minks maintained in dense populations [2, 3]. Lethal viruses, such as those in the USA and Denmark [6, 7], eventually die with the host. Therefore, a pathogenicity that is excessively strong becomes a selective pressure. This could be the reason why adaptation was not observed among mink-virus in Denmark, where only limited variants were found. This pressure is not as effective for humans as it is for minks, as some patients do not show symptoms. However, because most farmed minks are of the same age and have similar genetic backgrounds, the disease status can be

expected to be fairly uniform. Minks showed limited pathological traits in some countries [8], which suggested that this selection and attenuation is progressing within the mink population, or attenuated variants of human-derived virus are becoming apparent in minks.

The mink-derived human-virus in the Netherlands differ by only eight amino acids and 14 nucleotides from the closest human-virus. Although the mink-derived variant disappeared in Europe, if the variant remained in humans, the difference would not be safe enough for reverse mutations to re-adapt to humans.

Mutations of the group 2 mink-virus (Fig 1B) appeared to adapt to the new host, minks. Accumulating mutations will further reduce infectivity to humans. Variations among these strains can be increased by mink farming. Such strains could be maintained in minks or in Vero cells; among them, we could identify a combination of a strain and dosage that infects, but does not cause symptoms in humans. If this is achieved, vaccination will become possible by infecting the intestinal tract via oral administration. Therefore, local governments should encourage farmers to maintain their minks, rather than culling them. However, as minks can also carry human-virus as is, sequence analysis would be required to avoid this risk. It may be advisable to artificially infect minks with a known safe strain to restart farming. Human SARS-CoV-2 variants continue to change independently in each region [19], as shown in Fig 2A. Although some of the mink-virus from Denmark appear to be highly toxic to minks [7], the same group 1 variants would be the most adapted to humans, and they continue to mutate among humans in regions such as South Africa and Brazil. Some of these may have lower toxicity in minks.

## Supporting information

**S1 Fig. Situation in Denmark. A**. Number of confirmed cases, deaths, and PCs for samples. The blue arrow indicates the mink-derived human-virus (sPC8), which are far fewer than those in the Netherlands. **B**. Fatality rate (the number of deaths in the following week/the number of cases, grey) and percentage of mink-derived human-virus (red). The fatality rate remained fairly constant after the first wave subsided. **C**. The estimated confidence intervals confirmed the constancy of fatality.
(PNG)

**S1 Table. Rate of differences between mink-derived human-virus.** The differences between each variant in 1000 amino acid residues (positions are shown in Fig 1B and 1C by the IDs) and NB-EMC-35-3 | EPI_ISL_577774 (a mink-virus considered to be the same as that of humans). The rates of missense mutations are also shown. *Humans: differences between the mink-derived variants and similar human-virus in group 2, the Netherlands. The mink-derived human-virus were NB-EMC-45-4, NB-EMC-39-5, NB-EMC-7-2, NB-EMC-26-1, NB-EMC-45-3, NB-EMC-39-3, and NB-EMC-41-4. The related variants were NB-EMC-312, ZH-EMC-379, ZH-EMC-844, and ZH-EMC-845. Differences between averages were used.
(PDF)

## Acknowledgments

We would like to thank Editage (www.editage.com) for English language editing.

## Author Contributions

**Conceptualization:** Tomokazu Konishi.

**Data curation:** Tomokazu Konishi.

**Formal analysis:** Tomokazu Konishi.

**Funding acquisition:** Tomokazu Konishi.

**Investigation:** Tomokazu Konishi.

**Methodology:** Tomokazu Konishi.

**Project administration:** Tomokazu Konishi.

**Resources:** Tomokazu Konishi.

**Software:** Tomokazu Konishi.

**Supervision:** Tomokazu Konishi.

**Validation:** Tomokazu Konishi.

**Visualization:** Tomokazu Konishi.

**Writing – original draft:** Tomokazu Konishi.

**Writing – review & editing:** Tomokazu Konishi.

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
