## [Decision Letter · Decision Letter 0]

6 Mar 2021

PONE-D-21-04793

SARS-CoV-2 mutations among minks show reduced lethality and infectivity to humans

PLOS ONE

Dear Dr. Konishi,

Both reviewers like your paper and they have asked for some clarifications. Please respond and return your paper as soon as possible.

After careful consideration, we feel that it has merit but does not fully meet PLOS ONE’s publication criteria as it currently stands. Therefore, we invite you to submit a revised version of the manuscript that addresses the points raised during the review process.

We look forward to receiving your revised manuscript.

Kind regards,

Dong-Yan Jin

Academic Editor

PLOS ONE

Journal Requirements:

2.We suggest you thoroughly copyedit your manuscript for language usage, spelling, and grammar. If you do not know anyone who can help you do this, you may wish to consider employing a professional scientific editing service.  

Reviewers' comments:

Reviewer's Responses to Questions

**Comments to the Author**

1. Is the manuscript technically sound, and do the data support the conclusions?

Reviewer #1: Yes

Reviewer #2: Yes

2. Has the statistical analysis been performed appropriately and rigorously? 

Reviewer #1: Yes

Reviewer #2: Yes

3. Have the authors made all data underlying the findings in their manuscript fully available?

Reviewer #1: Yes

Reviewer #2: Yes

4. Is the manuscript presented in an intelligible fashion and written in standard English?

Reviewer #1: Yes

Reviewer #2: Yes

5. Review Comments to the Author

Reviewer #1: The manuscript is well-written. The authors use prinicipal component analysis and show that the evolution of human- and mink SARS-CoV-2 differs completely. In addition real-life data from the Netherlands and Denmark indicate that mink variants are less lethal and infective.

The authors suggest that mink-variants may be used for development of vaccines and therfore advise against culling of mink.

The findings in this paper is very important, but also controversial and political hot stuff. In Denmark it is still debated whether or not the culling of 17 millions mink was the right decision.

Reviewer #2: This is a nice and informative study. I have few comments and I request the author to clarify the comments.

1) It's not clear how the author derived the following results "Some mink-derived variants infected humans,

which accounted for 40% of the total SARS-CoV-2 cases in the Netherlands". The author presented this result in the "Introduction" of the main manuscript and then presented these as a part of results in the Abstract. Reading the article -- I didn't understand how the author derived the figure. Please clarify this.

2) In method the author mentioned "The axes were identified using 103 mink-virus and 6092 human-virus that were

proportionally selected from each continent. These data may be comparable to 130

million animals; since the mink population is estimated to be 50 million... ,".

How 103 minus 6092 human were comparable to 130 million animals? What does 130 million animals refers?

3) The author used abbreviation of worlds which need elaborated especially in figure legends and methods which now stands alone.

6. PLOS authors have the option to publish the peer review history of their article (what does this mean?). If published, this will include your full peer review and any attached files.

Reviewer #1: **Yes: **Carsten Schade Larsen

Reviewer #2: **Yes: **Dr. Najmul Haider, Postdoctoral Researcher, Royal Veterinary College, United Kingdom

---

## [Author Response · Author response to Decision Letter 0]

13 Mar 2021

Reviewer #1: The manuscript is well-written. The authors use prinicipal component analysis and show that the evolution of human- and mink SARS-CoV-2 differs completely. In addition real-life data from the Netherlands and Denmark indicate that mink variants are less lethal and infective.

The authors suggest that mink-variants may be used for development of vaccines and therfore advise against culling of mink.

The findings in this paper is very important, but also controversial and political hot stuff. In Denmark it is still debated whether or not the culling of 17 millions mink was the right decision.

I appreciate this comment. I tried to contact Danish officials. However, mink farming was banned in the country and the mink population was culled. The variants that could have supplemented the deficient vaccine were destroyed.

Reviewer #2: This is a nice and informative study. I have few comments and I request the author to clarify the comments.

1) It's not clear how the author derived the following results "Some mink-derived variants infected humans, which accounted for 40% of the total SARS-CoV-2 cases in the Netherlands". The author presented this result in the "Introduction" of the main manuscript and then presented these as a part of results in the Abstract. Reading the article -- I didn't understand how the author derived the figure. Please clarify this.

I rewrote the paragraph adding explanations concerning Fig. 1C, especially for the group indicated by the circle in the figure. I hope this improves the readability of the passage.

2) In method the author mentioned "The axes were identified using 103 mink-virus and 6092 human-virus that were

proportionally selected from each continent. These data may be comparable to 130

million animals; since the mink population is estimated to be 50 million... ,".

How 103 minus 6092 human were comparable to 130 million animals? What does 130 million animals refers?

I regret this shortage. I added an explanation by using an equation.

3) The author used abbreviation of worlds which need elaborated especially in figure legends and methods which now stands alone.

I appreciate the comment. The revised version includes the Materials and Methods section. A summary of all the abbreviations is included. Finally, I tried to find a replacement for DECIPHER, but it seems that this is rather a unique noun.

---

## [Editor Report · Decision Letter 1]

16 Mar 2021

SARS-CoV-2 mutations among minks show reduced lethality and infectivity to humans

PONE-D-21-04793R1

Dear Dr. Konishi,

We’re pleased to inform you that your manuscript has been judged scientifically suitable for publication and will be formally accepted for publication once it meets all outstanding technical requirements.

Kind regards,

Dong-Yan Jin

Academic Editor

PLOS ONE
---

## [Editor Report · Acceptance letter]

12 May 2021

PONE-D-21-04793R1 

SARS-CoV-2 mutations among minks show reduced lethality and infectivity to humans 

Dear Dr. Konishi:

I'm pleased to inform you that your manuscript has been deemed suitable for publication in PLOS ONE. Congratulations! Your manuscript is now with our production department. 

Kind regards, 

on behalf of

Professor Dong-Yan Jin 

Academic Editor

PLOS ONE